# Phytochemical Screening, Antioxidant and Sperm Viability of *Nelumbo nucifera* Petal Extracts

**DOI:** 10.3390/plants10071375

**Published:** 2021-07-05

**Authors:** Jiraporn Laoung-on, Churdsak Jaikang, Kanokporn Saenphet, Paiwan Sudwan

**Affiliations:** 1Department of Anatomy, Faculty of Medicine, Chiang Mai University, Chiang Mai 50200, Thailand; jiraporn_laoung@cmu.ac.th; 2Graduate School, Doctor of Philosophy Program in Anatomy, Faculty of Medicine, Chiang Mai University, Chiang Mai 50200, Thailand; 3Toxicology Section, Department of Forensic Medicine, Faculty of Medicine, Chiang Mai University, Chiang Mai 50200, Thailand; churdsak.j@cmu.ac.th; 4Department of Biology, Faculty of Science, Chiang Mai University, Chiang Mai 50200, Thailand; kanokporn.saenphet@cmu.ac.th

**Keywords:** oxidative stress, antioxidant, *Nelumbo nucifera* petal, sperm viability, agriculture waste

## Abstract

Sacred lotus **(***Nelumbo nucifera* Gaertn.; *N. nucifera***)** is a common ingredient in traditional medicine and Thai recipes. Its petal is an agricultural waste from stamen production. There are limitations in the used and pharmacological data of the petals resulting in more petals waste. The aims of this study were to investigate the phytochemical contents, antioxidant activity, and potential effects on sperm viability of aqueous (NAE) and ethanolic extracts (NEE) of both red and white *N*. *nucifera* petals. The white NAE had the highest total phenolics content, total tannins content and maximal antioxidant activity. The white NEE had the highest concentration of total flavonoids. Quercetin was a major flavonoid and was found in the aqueous extracts. Both red and white of NAE in the range of 0.22 to 1.76 mg/mL increased sperm viability. The white NAE was prominent in phytochemical content, antioxidant activity, and both red and white NAE effectively increased rat sperm viability in the in vitro model. The white NAE enhanced sperm viability by decreasing oxidative stress. It might be suggested that the *N. nucifera* petals have benefits for sperm viability health promotion and may increase the economic value of agricultural waste.

## 1. Introduction

Sacred lotus (*Nelumbo nucifera* Gaertn.; *N. nucifera*) is an aquatic perennial plant and belong to the Nelumbonaceae family [1]. It is reported that this plant as a medicinal plant has been used in traditional medicine [2]. Ayurvedic traditional medicine reported that most parts of *N**. Nucifera*, including the roots, stems, tender leaves, fruits, stamen, and flowers, can prevent and cure various diseases [3]. The previous study reported that the male rats that received 50% ethanol extract with *N. nucifera* seeds showed significant decreases in reproductive organ weight, testosterone level, sperm density, sperm motility, mount frequency, and intromission frequency [4,5]. However, the stamens of white *N. nucifera* known as the Siddha drug Venthamarai magarantha chooranam could enhance libido, which increased mount frequency and intromission frequency in male rats [6]. In addition, the study of *N. nucifera* petal tea had significantly increased rat sperm viability when compared with the control group [7]. Due to the National List of an Essential Medicine, Thailand, recognizing only the stamen of the red long flower of *N. nucifera* as a traditional drug [8], the stamens have high economic demand, and the petals are regarded as more agriculturale waste. The screening method reported the petals of *N. nucifera* contained essential phytochemical components, including phenols, flavonoids, tannins, alkaloids, saponin, steroid, terpenoids, cardiac glycosides, coumarin and quinone [9]. Phenols and flavonoids are antioxidants and can increase male reproductive ability [10]. Nevertheless, the petals of *N. nucifera* had little scientific research reported on male reproduction.

Male reproductive abnormality is one cause of infertility and has a strong impact on population growth rates [11]. Cell damage due to imbalance of free radicals or reactive oxygen species (ROS), free radical elimination, level of antioxidant capacity is an important cause of male infertility [12]. The iron (II) sulfate is a compound of iron that could be produced oxidative stress via superoxide radical formation [13]. Smoking, alcohol drinking and junk food are causes of oxidative stress formation in the human body and oxidative stress effect on the male reproductive system [14,15]. Moreover, lipid peroxidation, protein oxidation and DNA damage abnormally induce spermatogenesis, sperm concentration, sperm motility, sperm viability and sperm morphology [12,14,16,17,18]. These sperm qualities affect male fertility potential [16,17]. Oxidative stress also decreases the physiological capacity of fertilization, including capacitation, acrosome reaction, hyperactivation and sperm-oocyte binding, resulting in male infertility [14]

Male fertility potential can be enhanced by the prevention and management of oxidative stress. The amount of antioxidants consumption is a major factor to prevent oxidative stress from occurring [14]. Although synthetic antioxidants are easier to use and more effective, they are more expensive and have significant side effects [19]. These significant side effects are to affect DNA synthesis and cause carcinogen, fibrosis and embryotoxicity [20]. Therefore, the natural antioxidants have been extracted, purified and isolated from plants and have been used for the prevention and management of oxidative stress [12,14].

In Thailand, red and white *N. nucifera* are popularly grown for traditional Thai recipes, and more data exists for the red *N. nucifera* petals. Consequently, the aim of this studies to investigate the phytochemical content, antioxidant properties and effects on rat sperm viability of both white and red *N. nucifera* petals.

## 2. Results

### 2.1. The Total Phenolic, Total Flavonoid, Total Tannins, Total Monomeric Anthocyanins and Lycopene Content

The total amount of phytochemical contents of *N. nucifera* petals extracted by aqueous (NAE) and 95% ethanol (NEE) are presented in Table 1. The white NAE significantly had the highest total phenolic content followed by the red NAE, the white NEE extract and the red NEE with the presented values 24.15 ± 1.32, 20.67 ± 1.65, 0.67 ± 0.45, and 0.37 ± 0.44 µg GAE/g plant dried weight, respectively. The white NAE significantly had the highest total tannins content followed by the red NAE, the white NEE and the red NEE with the presented value 20.52 ± 1.07, 17.16 ± 3.01, 1.43 ± 0.49 and 1.05 ± 0.21 µg TAE/g plant dried weight, respectively. Total flavonoids content significantly showed the highest in the white NEE (21.59 ± 1.00 µg QE/g plant dried weight), followed by the red NEE, the red NAE and the white NAE (19.74 ± 0.77, 10.61 ± 1.25 and 7.83 ± 0.37 µg QE/g plant dried weight, respectively). The red NAE showed only total monomeric anthocyanins. However, the lycopene content of the red *N. nucifera* petals (0.35 ± 1.47 µg/mg plant dried weight) presented no significant difference when compared with the white *N. nucifera* petals (0.88 ± 1.16 µg/mg plant dried weight).

### 2.2. Antioxidant Properties

The antioxidant properties of *N. nucifera* petals extracted by aqueous and 95% ethanol are presented in Table 2.

The aqueous and ethanolic extract of both red and white *N. nucifera* petals scavenged DPPH radical in a dose-dependent manner, and the results are shown in Figure 1. The aqueous extracts and gallic acid (used as a standard) had potential DPPH radical scavenging more than the ethanolic extracts. The half-maximal inhibitory concentration (IC50) of DPPH radical scavenging of the white NAE, the red NAE, gallic acid, the white NEE and the red NEE presented 13.31 ± 1.96, 14.60 ± 1.55, 14.95 ± 0.18, 481.41 ± 18.53 and 634.79 ± 21.65 µg/mL, respectively.

The aqueous and ethanolic extract of both red and white *N. nucifera* petals scavenged ABTS radical in a dose-dependent manner, and the results are shown in Figure 2. More potential was seen in the gallic acid ABTS radical scavenging in the aqueous extracts and the ethanolic extracts. However, the aqueous extracts had potential ABTS radical scavenging more than the ethanolic extracts. The half-maximal inhibitory concentration (IC50) of ABTS radical scavenging of gallic acid, the white NAE, the red NAE, the white NEE and the red NEE, presented 0.45 ± 0.03, 10.95 ± 0.56, 13.47 ± 2.58, 127.45 ± 22.84 and 173.89 ± 25.52 µg/mL, respectively.

The aqueous and ethanolic extract of both red and white *N**. nucefera* petals scavenged H_2_O_2_ in a dose-dependent manner, and the results are shown in Figure 3. The aqueous and ethanolic extracts exhibit greater potential of H_2_O_2_ radical scavenging compared to gallic acid. The half-maximal inhibitory concentration (IC50) of H_2_O_2_ radical scavenging of the white NAE, the red NAE, the white NEE, the red NEE and gallic acid presented 134.02 ± 2.29, 160.38 ± 6.72, 1014.98 ± 16.94, 1391.22 ± 62.99 and 1965.88 ± 4.48, respectively.

The aqueous and ethanolic extracts of both red and white *N**. nucefera* petals reduced Fe (III) to Fe (II) in a dose-dependent manner, and the results are shown in Figure 4. The aqueous extracts and gallic acid had the potential for reducing power more than the ethanolic extracts. The half-maximal inhibitory concentration (IC50) of reducing power of the white NAE, the red NAE, gallic acid, the red NEE and the white NEE presented 59.30 ± 11.08, 90.84 ± 12.34, 224.91 ± 3.14, 398.75 ± 43.13 and 483.85 ± 3.14, respectively.

### 2.3. Lipid Peroxidation Assay

The aqueous and ethanolic extract of both red and white *N**. nucefera* petals inhibited LPO, which was induced by FeSO_4_ in the linoleic acid model. The percent inhibitions are shown in Figure 5 and indicate that the plant extracts inhibited the LPO formation. More potential LPO inhibition was seen in the aqueous extracts and gallic acid than in the ethanolic extracts. The concentration of the half-maximal inhibitory concentration (IC50) values of LPO inhibition of the white NAE, the red NAE, gallic acid, the red NEE and the white NEE showed 0.05 ± 0.01, 0.10 ± 0.01, 0.11 ± 0.11, 0.17 ± 0.10 and 1.35 ± 0.60 µg/mL, respectively (Table 3).

### 2.4. Advance Oxidation Protein Products (AOPP) Assay

The aqueous and ethanolic extracts of both red and white *N**. nucefera* petals inhibited AOPP, which was induced by FeSO_4_ in the BSA model. The percent inhibitions are shown in Figure 6 and indicate that the plant extracts inhibited AOPP formation. The aqueous extracts and gallic acid increased the potential AOPP inhibition compared to the ethanolic extracts. The concentration of the half-maximal inhibitory concentration (IC50) values of AOPP inhibition of the white NAE, the red NAE, gallic acid, the red NEE and the white NEE showed 0.35 ± 0.07, 2.66 ± 0.49, 4.21 ± 0.07, 11.88 ± 0.80 and 27.19 ± 6.00 µg/mL, respectively (Table 3).

### 2.5. Advance Glycation End Products (AGEs) Assay

The aqueous and ethanolic extract of both red and white *N**. nucefera* petals inhibited AGEs, which was induced by D-glucose in the BSA model. The percent inhibitions are shown in Figure 7 and imply hat the plant extracts inhibited

AGEs’ formation. However, the aqueous extracts, the ethanolic extracts and gallic acid had no significant difference in the potential in IC50 values. The concentration of the half-maximal inhibitory concentration (IC50) values of AGEs’ formation of the red NAE, the red NEE, gallic acid, the white NAE and the white NEE showed 0.55 ± 0.24, 1.00 ± 0.70, 1.05 ± 0.12, 1.07 ± 0.93 and 1.23 ± 1.35 µg/mL, respectively (Table 3).

### 2.6. High-Performance Liquid Chromatography (HPLC) Analysis

From the HPLC screening, phenolics (250 nm) showed in the red NAE (A), the white NAE (B), the red NEE (C) and the white NEE (D). For flavonoids (330 nm) in the red NAE (E), the white NAE (F), the red NEE (G) and the white NEE (H), they were adequately separated within 60 min (Figure 8). The retention time and standard UV spectrum of the phytochemical standard were used for identification. Quercetin, gallic acid, catechin and *p*-hydroxybenzoic acid were found in the *N. nucifera* petal extracts, and the retention times were 2.464, 4.356, 10.905 and 20.821 min, respectively.

### 2.7. Sperm Viability

Since the antioxidant properties present in the NAE were higher than in the NEE, the NAE was applied for sperm viability screening. The result showed the red NAE and the white NAE can preserve rat sperm viability. The percentages of sperm viability are shown in Figure 9 and indicate that the plant extracts can preserve or promote sperm viability in the in vitro model. The dose of red NAE and white NAE in the range of 0.22–1.76 mg/mL significantly increased sperm viability when compared to the control group. Conversely, the dose of 3.52 mg/mL in both red and white NAE showed no difference when compared to the control group. Sperm viabilities were classified with DAPI (4′,6-Diamidino-2-Phenylindole, Dihydrochloride) and PI (Propidium iodide) and investigated under ImageXpress Micro 4 High-Content Imaging System (Figure 10).

### 2.8. Antioxidant Properties in Sperm

The white NAE presented the most effective antioxidant properties in a cell-free system. Therefore, white NAE was applied in order to test antioxidant properties in sperm.

Each sperm suspension was treated with various concentrations of white NAE and FeSO_4_ was added to induce AOPP and additional antioxidant statuses. The white NAE were thus designed to test for potential scavenging activity. In the supernatant, AOPP levels were measured. The result showed non-significant changes in the normal control and all doses of white NAE, while the level of AOPP significantly increases in the supernatant with FeSO_4_ when compared with the normal control and all doses of white NAE (Figure 11). The AOPP levels in the groups treated with FeSO_4_ plus white NAE in the dose of 0.44 mg/mL was significantly lower than the FeSO_4_ group and was similar to the levels measured in the normal control. However, the supernatant withFeSO_4_ plus white NAE in the doses of 0.22, 0.88, 1.76 and 3.52 mg/mL was not significantly different in the FeSO_4_ groups (Figure 11).

The result of the total oxidative status in all doses of white NAE’s supernatants were the same as those of the AOPP. The level of total oxidative status significantly increases in FeSO_4_ when compared with the normal control and all dose of white NAE’s supernatants. The FeSO_4_ plus white NAE in the doses of 0.22 and 0.44 mg/mL were significantly lower than the FeSO_4_ group and were similar to the levels measured in the normal control group (Figure 12).

The total antioxidant status was shown in Figure 13, which shows a significant increase in the white NAE at the doses of 0.44, 0.88, 1.76 and 3.52 mg/mL when compared to the normal control group. All groups treated with FeSO_4_ plus white NAE showed a non-significant increase when compared to the FeSO_4_ group, except the dose of 1.76 and 3.52 mg/mL were significantly higher than the FeSO_4_ group. The LPO and AGEs could not be detected in this rat sperm model.

Additionally, there were certain effects on sperm from this experiment. The percentage of the sperm viability using trypan blue staining showed 91.39 ± 1.98 (control), 90.5 ± 1.06 (0.22 mg/mL), 91.39 ± 1.85 (0.44 mg/mL), 87.94 ± 1.51 (0.88 mg/mL), 80.56 ± 1.45 (1.76 mg/mL), 75.61 ± 0.82 (3.52 mg/mL), 19.39 ± 6.78 (FeSO_4_), 87.44 ± 0.92 (FeSO_4_ + 0.22 mg/mL), 90 ± 0.61 (FeSO_4_ + 0.44 mg/mL), 79.78 ± 1.56 (FeSO_4_ + 0.88 mg/mL), 65.72 ± 2.61 (FeSO_4_ + 1.76 mg/mL) and 33.44 ± 1.83 (FeSO_4_ + 3.52 mg/mL). The results showed non-significant changes in the normal control group and the white NAE group at the doses of 0.22, 0.44 and 0.88 mg/mL. In contrast, the percentage of sperm viability significantly decreased in the white NAE at the dose of 1.76 and 3.52 mg/mL and FeSO_4_ when compared with the normal control group. The percentage of viable sperm in the groups treated with FeSO_4_ plus white NAE was significantly higher than the FeSO_4_ group. The FeSO_4_ plus white NAE group at the doses of 0.22 and 0.44 mg/mL were similar to the percentage in the normal control group. The result showed consistency in the DAPI and PI staining. However, the percentage of the sperm viability was non-significantly different between the group treated with white NAE at doses of 0.22 and 0.44 mg/mL and the normal control group. This evidence may affect the duration of the incubation period resulting in different sperm viability in trypan blue and DAPI and PI staining.

## 3. Discussion

The white NAE contained the highest amounts of phenolic compounds, followed by the red NAE, the white NEE and the red NEE. The results showed that the aqueous extracts are composed of more phenolic compounds than ethanolic extracts. Sugars are also extracted with water from plant material. Reducing sugars are generally known to react with Folin-Ciocalteu reagent and might impact the results of the total phenolics [21]. This means that components of *N. nucifera* petals phenolics are polar because of water being the polar solvent rather than 95% ethanol [22,23]. Thus, solvent polarity will play an important role in increasing phenolic solubility [24]. The total tannins content of the aqueous and ethanolic extract of white and red *N. nucifera* petals presented similar variations with the results of the total phenolics content because tannins are phenolic compounds found in plants that are soluble in water and polar organic solvents [25]. The result shows the aqueous extracts contain more tannins than ethanolic extracts. This report indicated that most components in *N. nucifera* petals tannins are polar tannins because of water being the polar solvents than 95% ethanol [26].

The result of the total flavonoids content of the aqueous and ethanolic extracts of white and red *N. nucifera* petals showed that the white NEE contained the highest amount of flavonoids content, followed by the red NEE. Moreover, the ethanolic extracts had a significantly higher flavonoids content than the aqueous extracts. Similarly, the absolute ethanolic extract of *Eucalyptus pyriformis* found more quercetin than when extracted by distilled water [22]. Quercetin is a typical flavonoid present in fruits and vegetables [27]. Most flavonoids are less polar or semi-polar [23]. This means that most of the flavonoids in *N**. nucefera* petals may be less polar or semi-polar.

The red NAE showed only total monomeric anthocyanins. Anthocyanins are water-soluble vacuolar plant pigments responsible for the bright colors red, purple or blue of flowers, skin, seeds, fruits and leaves. It is obvious that a sugar unit of anthocyanin increases the solubility of these molecules in water. Therefore, anthocyanins are not found in white *N. nucifera* petals, and the red NEE might be oxidized by temperature and light from the steamed, dried in an oven and extracted method, resulting in a color change and degradation [28]. In addition, it had been reported that hot water is an organic solvent that the mostsuitable solvent for polyphenols extraction from the plant sample and this solvent is the most used method for water-soluble phytochemicals from crude drugs [29]. This proper method was used in this experiment that is common Thai recipes and traditional medicine.

The HPLC of the white NAE, the red NAE, the white NEE, and the red NEE showed quercetin, gallic acids, catechin and *p*-hydroxybenzoic acid, which the flavonoids and phenolics content. Consistently, the literature presented quercetin as the major substance in the lotus, followed by hydroxycinnamic acid and hydroxybenzoic acid [30]. This study demonstrates aqueous extraction as a good alternative to extract the polyphenol compounds from *N. nucifera* petals.

The antioxidant activity of the aqueous and ethanolic extract of white and red *N. nucifera* petals was evaluated by the DPPH radical scavenging, ABTS radical scavenging, H_2_O_2_ scavenging and reducing power of Fe^3^^+^. The white NAE presented the most effective for DPPH radical scavenging, ABTS radical scavenging, H_2_O_2_ scavenging and reducing power of Fe^3^^+^, followed by the red NAE. According to the amount of polyphenol compounds, including phenolics, tannins, flavonoids, anthocyanins and lycopene, they actas antioxidant compounds. A DPPH radical scavenging, ABTS radical scavenging, and H_2_O_2_ scavenging assays are based on a hydrogen donor for the radical scavenging or antioxidant reaction. The color of these reactions is changed according to the amount of phytochemical contents in the extracts [31,32]. In addition, the reducing power is based on a single electron transfer mechanism, in which Fe (III) is reduced to Fe (II) [23]. The white NAE had the highest antioxidant capacity because it contained the highest amounts of polyphenols and may be the result of a hydrogen donor or a single electron transfer.

Lipids, proteins and carbohydrates are the macromolecules in living cells and free radical targets [14]. The linoleic acid is the source of lipid for in vitro study lipid peroxidation (LPO) induced by FeSO_4__._ Linoleic acid is a highly unsaturated fatty acid in the cell membrane [15]. The white NAE showed a significantly low concentration of the half-maximal inhibitory concentration (IC50) values of LPO, followed by the red NAE, the red NEE and the white NEE.

Proteins are the most common component in living cells and are easy to oxidize [15] The bovine serum albumin is a source of protein for in vitro study protein oxidation induced by FeSO_4__._ The result of AOPP inhibition was similar to the LPO inhibition. The phenolics and flavonoids show free radical scavenging ability and prevent LPO, AOPP and AGEs formation [22,33], resulting in the prevention of cellular damage [12]. The white NAE had the highest phenolic compounds, leading to a strong antioxidant activity and showed significantly decreased LPO and AOPP levels.

Normally in Thai recipes and traditional medicine, *N. nucifera* petals are consumed directly or cooked with water, and in the present study, the petals presented high antioxidant properties in the NAE. The aqueous extracts were screened for the effect of both the red and white NAE on sperm viability in an in vitro model. The results showed both red and white NAE significantly increased sperm viability when compared with the control group. After semen ejaculation, the sperm had oxidative stress due to the inappropriate environment, leading to sperm cell damage and causing sperm death [34]. Abnormal semen samples treated with 5′-N-ethyl-carboxamidoadenosine (NECA), an antioxidant agent, prevented oxidative stress in the sperm, thereby preserving sperm viability [35]. In this study, both red NAE and white NAE promoted sperm viability during periods of oxidative stress, indicating the phytochemical contents in *N. nucifera* petals might act directly as a free radical scavenger. Moreover, NAE may inhibit the formation of LPO, AOPP and AGEs. Appropriate concentrations, ranging from 0.22–1.76 mg/mL, showed normal morphology. This effect may be similar to a previous in vitro study of *N. nucifera* petal tea, which increased sperm viability when compared with the control group [7]. and the result had a positive effect on male reproduction, similarly to how the stamen of *N. nucifera* could enhance male sexual behavior [6]. Conversely, *N**. Nucifera*’s seed had an antifertility effect on male reproduction [4,5]. In addition, this experiment was designed for FeSO_4_-induced AOPP and total oxidative status in the rat’s sperm solutions, and the white NAE had a beneficial effect on scavenging free radicals in the rat sperm that were induced by FeSO_4_, resulting in the prevention of cellular damage [12]. The study showed that the appropriate concentrations of 0.22 and 0.44 mg/mL showed lower AOPP and total oxidative levels and were similar to the normal control group. Furthermore, the white NAE promoted sperm viability via trypan blue during oxidative stress, in which the phytochemical contents in this extract might be directly acting as a free radical scavenger. Although this study showed positive results from using the extracts for sperm viability, it is possible that sperm motility and sperm morphology may take place, which may affect sperm fertilization. This will need to be investigated in further studies.

## 4. Materials and Methods

### 4.1. Chemicals and Reagents

Folin-Ciocalteu reagent, potassium acetate, sodium acetate, 2,2-diphenyl-1-Picrylhydrazyl (DPPH), 2,2′-Azino-di-(3-Ethylbenzthiazoline Sulfonic acid) (ABTS), Potassium hexacyanoferrate, thiobarbituric acid, xylenol orange, quinine hemisulfate and chloramine-T were purchased from Sigma-Aldrich (St. Louis, MO, USA). Acetic acid (glacial) and formic acid were purchased from Merck KGaA (Darmstadt, Germany). Phytochemical standards, including quercetin, gallic acid, catechin and *p*-hydroxybenzoic acid, were purchased from Sigma-Aldrich (St. Louis, MO, USA).

### 4.2. Plant Collection and Extraction

Both red and white petals of *N. nucifera* were collected from the Thung Yang subdistrict, Laplae district, Uttaradit Province, Thailand (17°31′09.7″ N, 99°59′01.6″ E), in September 2019. The samples were deposited and authenticated at Herbarium, Faculty of Pharmacy, Chiang Mai University, voucher number is 023248-1 and 023248-2, respectively. The petals were washed, steamed and dried at 60 °C. The dried petals were pulverized and stored at 4 °C before use. The dried petals were extracted one time with hot distilled water at 75–80 °C (The aqueous extract of *N. nucifera* petals: NAE) and 95% ethanol (The ethanolic extract of *N. nucifera* petals: NEE). Then, the solutions were filtered with a syringe filter and diluted with extract solution before experimentation. The concentration of final stock solutions is 100 mg/mL. The research activities were shown in schematic diagram (Figure 14).

### 4.3. Total Phenolic Contents

The total phenolic content of the *N. nucifera* petals using a previously reported method [36]. First, 50 µL of the sample and 500 µL of 1 M Sodium carbonate (Na_2_CO_3_) were added into 500 µL of 10% Folin-Ciocalteu reagent. The solution was incubated for 15 min at room temperature and measured at 765 nm by spectrophotometer (Shimadzu UV-2401PC Thermo Fisher Scientific, Waltham, MA, USA). The total phenolic content was expressed in µg of gallic acid equivalents per g plant dried weight.

### 4.4. Total Flavonoids Contents

The total flavonoids contents were determined using colorimetric assay [36]. Briefly, 50 µL of 10% aluminum chloride was added to 100 µL of the plant extracts and incubated at room temperature for 30 min. Then, 50 µL of potassium acetate and 700 µL of distilled water were added. The absorbance was measured at 415 nm by a spectrophotometer. The total flavonoid content was expressed in µg of quercetin equivalents per g plant dried weight.

### 4.5. Total Tannins Content

The Folin-Ciocalteu method was carried out to determine the total tannins content [36] in *N. nucifera* petal. Tannic acid was used as the standard for calibration, and the results were express as µg of tannic acid equivalents per g plant dried weight. A 50 µL sample was mixed with 500 µL of 10% Folin-Ciocalteu reagent and 400 µL of 1 M sodium carbonate (Na_2_CO_3_). The mixture solution was incubated for 30 min at room temperature. The absorbance was measured at 700 nm by a spectrophotometer [37].

### 4.6. Total Monomeric Anthocyanins

The total anthocyanin content was estimated by the pH-differential method, and the results were expressed as µg of cyanidin-3-glucoside equivalents per g plant dried weight [38]. Two dilutions were prepared, both with 0.025 M potassium chloride buffer (pH 1.0) and 0.4 M sodium acetate buffer (pH 4.5). A sample solution volume of 100 µL was added in a 5 mL tube with 900 µL of 0.025 M potassium chloride buffer (pH 1.0) or 0.4 M sodium acetate buffer (pH 4.5). These dilutions were incubated at room temperature for 15 min. The absorbance of each dilution was measured at 510 nm and 700 nm by spectrophotometer (Shimadzu UV-2401PC). The absorbance of the diluted sample (A) was calculated as follows:A = (A_510_ − A_700_)_pH 1__.0_ − (A_510_−A_700_)_pH 4__.5_

The monomeric anthocyanin concentration in the stock solution was calculated using the following formula: Monomeric anthocyanin pigment (µg/mL) = (A × MW × DF × 1000)/(ε × 1)

Anthocyanin content was calculated as cyanidin-3-glucoside, when MW = 449.2 and ε = 26,900.

### 4.7. Lycopene Content Analysis

Lycopene was extracted using a hexane: ethanol: acetone (2:1:1) (*v*/*v*) mixture. First, 1 g of each dried sample was dissolved in 1 mL DW, vortexed and incubated for 60 min at 30 °C in a water bath. Then, 8 mL of hexane: ethanol: acetone (2:1:1) were added, vortexed and incubated for 10 min in the dark. After incubation, 1 mL of DW water was added and vortexed. These supernatants were collected and measured at 503 nm by spectrophotometer (Shimadzu UV-2401PC Thermo Fisher Scientific, Waltham, MA, USA) [39]. The lycopene content was calculated according to:Lycopene (µg/mg plant dry weight) = A_503_ × 537 × 8 × 0.55/0.10 × 172

### 4.8. Antioxidant Properties

#### 4.8.1. 2,2-Diphenyl-1-Picrylhydrazyl (DPPH) Radical Scavenging Assay

DPPH radical scavenging assay was carried out to determine the free radical scavenging capacity in *N. nucifera* petal extracts [36]. A 100 µL of the various concentration of NAE and NEE were added into 1 mL of 0.004% DPPH solution in methanol. The reaction mixture was incubated for 30 min in the dark and measured at 515 nm by spectrophotometer (Shimadzu UV-2401PC). Quercetin was used for positive control, and the results were calculated and expressed in percentage of inhibition according to:% inhibition = (A_DPPH_ − A_sample_)/A_DPPH_ × 100

#### 4.8.2. 2,2′-Azino-di-[3-Ethylbenzthiazoline Sulfonate] (ABTS) Radical Scavenging Assay

ABTS radical cation decolorization assay was determined the free radical scavenging activity of NAE and NEE [40]. ABTS was prepared and stored in the dark at 4 °C. The stock solution was diluted with distilled water to obtain an absorbance of 0.7 at 734 nm. Next, 200 µL of ABTS working solution was added in 50 µL of the various concentration of plant extract and incubated for 30 min in darkness. The absorbance was measured at 743 nm using a microplate reader (Bio Tek Synergy H4 Hybrid Microplate Reader, BioTek Instruments, Winooski, VT, USA), and gallic acid was used for positive control. The results were calculated and expressed in percentage of inhibition according to:% inhibition = (A_ABTS_ − A_sample_)/A_ABTS_ × 100

#### 4.8.3. Hydrogen Peroxide Scavenging Assay

The hydrogen peroxide (H_2_O_2_) scavenging activity of NAE and NEE was determined using modified the method of Fernando et al. [41]. Then, 2.5 mL of 40 mM H_2_O_2_ was added to 500 µL of the plant extract and incubated for 10 min at room temperature in darkness. The absorbance was measured at 743 nm by spectrophotometer (Thermo Scientific™ GENESYS™ 10S UV-Vis, Thermo Fisher Scientific, Waltham, MA, USA), and gallic acid was used for positive control. The results were calculated and expressed in percentage of inhibition according to:% inhibition = (A_H_2_O_2__ − A_sample_)/A_H_2_O_2__ × 100

#### 4.8.4. Reducing Power Assay

The reducing power activity of NAE and NEE was determined using spectrophotometric detection of the Fe (III)-Fe (II) reduction method, and the results were expressed as the absorbance at 700 nm and the IC50 value of Fe (III) reduction [42]. Briefly, 250 µL of phosphate-buffered saline (PBS) (0.2 M, pH 6.6) and 250 µL 1% potassium hexacyanoferrate (III) (K_3_Fe (CN)_6_) were added to a 5 mL tube with 20 µL of the plant extract, followed by incubation at 50 °C for 20 min. After incubation, 250 µL of 10% trichloroacetic acid (TCA) was added and centrifuged at 1000× *g* rpm for 10 min. Next, 250 µL of the supernatants were collected and added to 250 µL of distilled water and 500 µL of 1% ferric (III) chloride (FeCl_3_), respectively. The absorbance was measured at 700 nm by a spectrophotometer (Thermo Scientific™ GENESYS™ 10S UV-Vis).

### 4.9. Lipid Peroxidation Assay

Thiobarbituric acid-reactive species (TBARS) assay was carried out to determine the inhibition of lipid peroxidation of NAE and NEE [43]. Briefly, 50 µL of plant extract was added in the mixture of linoleic acid emulsion (125 µL, 10 mM in 1M PBS, pH: 7.4) and 12.5 µL of FeSO_4_ solution (0.07 M). The mixture volume was made up to 287.5 µL by distilled water, and the solution was incubated for 30 min at room temperature. A 225 µL of 0.85% normal saline solution, 500 µL 10% TCA, and 100 µL of thiobarbituric acid (TBA) was added before boiling at 95 °C for 30 min and followed by centrifugation at 3500× *g* rpm for 10 min. The absorbance of the supernatant was measured at 532 nm with a microplate reader. The percent of inhibition was calculated following the equation:% Inhibition = (OD_control_ − OD_sample_)/OD_control_ × 100

### 4.10. Inhibition of Advance Oxidation Protein Products (AOPP) Formation

The inhibition of AOPP of NAE and NEE using a previously reported method [44]. Briefly, 50 µL of plant extract was added in the bovine serum albumin (BSA) (135 µL, 1 mg/mL in 0.2 M PBS, pH: 7.4) and 15 µL of FeSO_4_ solution (0.07 M). The solution was mixed and incubated for 30 min in the darkness. After incubation, 50 µL of 1.16 M potassium iodide (KI) and 500 µL 10% TCA was added, followed by 30 min of incubation, and 20 µL of absolute acetic acid was added after incubation. A 200 µL of supernatant was added in 96 well-plate and measured at 340 nm by a microplate reader. The percent of inhibition was calculated following the equation:% Inhibition = (OD_control_ − OD_sample_)/OD_control_ × 100

### 4.11. Inhibition of Advance Glycation End Products (AGEs) Formation

The modification method from *Rashmi* et al. was using determined the anti-AGEs assay [45]. Briefly, 50 µL of plant extract was added into 96 well-plate with 50 µL of BSA (1 mg/mL in 0.2 M PBS, pH: 7.4) and 50 µL of d-glucose (1 M) and incubated at 37 °C for 24 h. The fluorescence intensity (excitation wavelength 360 nm and emission wavelength 460 nm were measured by a microplate reader. The percent of inhibition was calculated by the following equation:% Inhibition = (OD_control_ − OD_sample_)/OD_control_ × 100

### 4.12. Analysis of Phytochemical Content by High-Performance Liquid Chromatography (HPLC)

Phytochemical profiles of both red and white NAE and NEE were screened by high-performance liquid chromatography (HPLC)-diode array (Agilent 1260 Infinity Binary LC, Santa Clara, CA, USA) [46]. The HPLC condition comprised of Purospher^®^ Star PR-18 endcapped column (150 × 4.60, 5 µm). The mobile phase consists of 92% A (0.1% formic acid in water) and 8% B (acetonitrile), maintained for 10 min. The B was increased to 14% in 24 min, 23% in 35 min and 24% in 60 min. Then, 10 µL of the sample was injected, and the spectra were determined at 250 nm and 330 nm [47]. The spectra between 200 and 400 nm were collected. The identification of the chromatographic peak was achieved by comparing the retention times and spectral characteristics of the eluted peaks with the standards.

### 4.13. Sperm Viability

The sperm suspension prepared from collected right caudal epididymis of three normal mature male Wistar rats (*Rattus norvegicus*) aged 8–10 weeks and minced in 10 mL of Krebs–Henseleit solution. Then sperm concentration was investigated by hemocytometer. The experimental procedure was approved by the Animal Ethics Committee, Faculty of Medicine, Chiang Mai University (No. 44/2019).

The fresh sperm solution 100 µL of was added to 96 well-plates. The first control group added 100 µL of Krebs–Henseleit solution. The experimental groups were added at 100 µL of various concentrations of NAE in each well and incubated at 37 °C for 3 h. Then the mixture solution was stained with 5 µL of 5 µg/mL of propidium iodide (PI) solution, incubated at 37 °C for 15 min, followed by 10 µL of 1 µg/mL of 4′,6-Diamidino-2-Phenylindole, Dihydrochloride (DAPI). The sperm viability was determined at a magnification of 400× under ImageXpress Micro 4 High-Content Imaging System. The total sperms in two fields per well were counted classified by image J processing and analysis program [7]. 

### 4.14. Antioxidant Properties in Sperm

The fresh sperm solution was divided into twelve equal parts, and 450 µL of sperm solution was added to seperate test tubes. The first control group had 125 µL of Krebs–Henseleit solution added, whereas the second through sixth tubes were supplemented with five different concentrations of white NAE in each tube. For tubes 7-12, they were induced with 25 µL of FeSO_4_ and tubes 8-12 were each subsequently supplement with 100 µL of various concentrations of white NAE in each tube then incubated at 37 °C for 2 h. After incubation, the aliquots were centrifuged at 1755× *g* rpm for 5 min. The spermatozoa were stained with trypan blue, and the spermatozoa were counted and viability discriminated under 40 objective lens of a light microscope [48]. In addition, the supernatant was stored at -20 °C until the antioxidant properties were assessed.

#### 4.14.1. Lipid Peroxidation (LPO) Assay

Thiobarbituric acid-reactive species (TBARS) assay was carried out to determine the inhibition of lipid peroxidation [43]. In total, 100 µL supernatant was assessed by the determination of TBARS consistently the method in a cell-free system. The standard calibration curve was obtained using 1,1,3,3-tetramethoxypropane, and the result was expressed in mEq µmol 1,1,3,3-tetramethoxypropane/L.

#### 4.14.2. Inhibition of Advance Oxidation Protein Products (AOPP) Formation

In total, 100 µL of supernatant was assessed by the determination of AOPP followed the method in a cell-free system. Chloramine-T was used as the standard for calibration, and the result was expressed in mEq µmol chloramine-T/L.

#### 4.14.3. Inhibition of Advance Glycation End Products (AGEs) Formation

Total 100 µL of supernatant was added into a 96 well-plate and was assessed similarly in the cell-free system model. Quinine hemisulfate was used as the standard for calibration, and the result was expressed in mEq µmol quinine hemisulfate/L.

#### 4.14.4. Ferric-Xylenol Orange (FOX1) Assay

The total oxidant status was determined by the resultant of ferric ion and provided an indirect measure of the peroxide content. The FOX1 reagent was prepared by mixing 1 mL of ferrous sulfate solution (25 mM FeSO_4_ in 2.5 M H_2_SO_4_), 50 µL of xylenol orange solution (12.5 M xylenol orange in methanol), 90 mL of sorbitol solution (100 mM sorbitol in ultrapure water) and 8.95 mL of autoclaved double distilled water. Immediately afterward, 200 µL of FOX1 reagent was added into a 96 well-plate with 50 µL of supernatant and then incubated at room temperature for 15 min. The absorbance of the supernatant was measured at 560 nm by a microplate reader. Hydrogen peroxide was used as the standard for calibration, and the result was expressed in mEq µmol hydrogen peroxide/L [49].

#### 4.14.5. Ferric Reducing Antioxidant Power (FRAP) Assay

The total antioxidant status was evaluated by the ability of the samples to reduce ferric ions, which is measured as an absorbance change of the ferrous TPTZ complex. FRAP reagent was prepared by mixing 100 mL of acetate buffer (300 mM, pH 3.6), 10 mL of TPTZ solution (10 mM TPTZ in distilled water) and 10 mL of FeCl_3_ solution (20 mM FeCl_3_ in distilled water). Immediately after, 200 µL of the FRAP reagent was added into a 96 well-plate with 50 µL of supernatant and then incubated at 37 °C for 4 min. The absorbance of the supernatant was measured at 593 nm by a microplate reader. Gallic acid was used as the standard for calibration, and the result was expressed in mEq µmol gallic acid/L [50].

### 4.15. Statistical Analysis

The data showed the descriptive mean ± standard deviation (SD). The half-maximal inhibitory concentration (IC50) value was calculated using Excel Microsoft 365. The normal distribution was assessed via Kolmogorov-Smirnov test. Mean values of lycopene content, DPPH radical scavenging, ABTS radical scavenging, LPO, AGEs, sperm viability, AOPP in sperm and total oxidative status in sperm were statistically analyzed by a One-way ANOVA following with multiple comparisons by Tukey’s tests performed to analyze the differences between groups. Mean values of other parameters were analyzed using the Kruskal-Wallis, followed by Mann-Whitney tests, which were performed to analyze the differences between groups. All experiments were done in three replications, and the significance level was set at *p* < 0.05.

## 5. Conclusions

In conclusion, *N. nucifera* petal contained rich phenols and flavonoids and had potential in antioxidant activities in the cell-free system. The white NAE was the most effective in phytochemical content and antioxidant activity, and both red and white NAE effectively increased sperm viability in the in vitro model. The white NAE enhanced sperm viability by decreasing oxidative stress and increasing antioxidants. From these results, it may be suggested that the *N. nucifera* petals had benefits for health promotion and increasing the economic value of agricultural waste. Moreover, the effect of white NAE on antioxidant activity and male reproductive function in animals and humans should be further studied.

## Figures and Tables

**Figure 1 plants-10-01375-f001:**
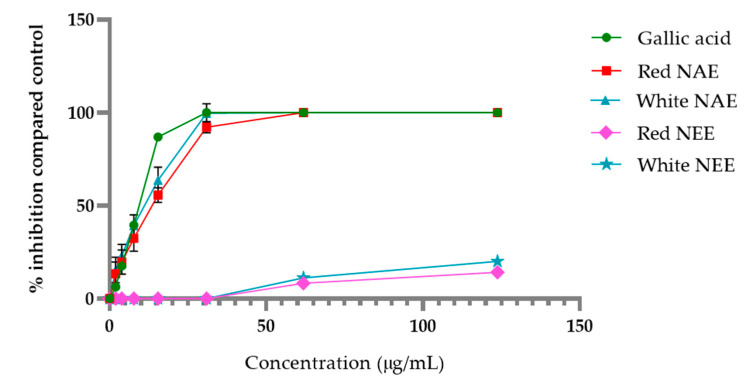
The percent of inhibition of DPPH of both aqueous and ethanolic extract of red *N. nucifera* petals and white *N. nucifera* petals. (NAE: *N. nucifera* petals aqueous extraction, NEE: *N. nucifera* petals ethanolic extraction). Data are mean values ± standard deviation (error bars).

**Figure 2 plants-10-01375-f002:**
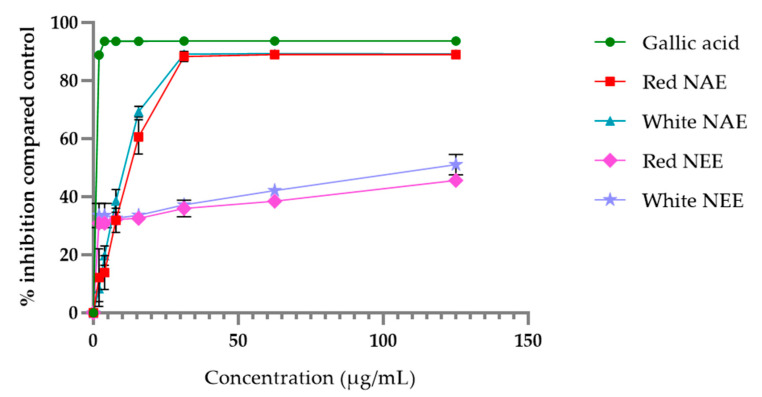
The percent of inhibition of ABTS of both aqueous and ethanolic extracts of red *N. nucifera* petals and white *N. nucifera* petals. (NAE: *N. nucifera* petals aqueous extraction, NEE: *N. nucifera* petals ethanolic extraction). Data are mean values ± standard deviation (error bars).

**Figure 3 plants-10-01375-f003:**
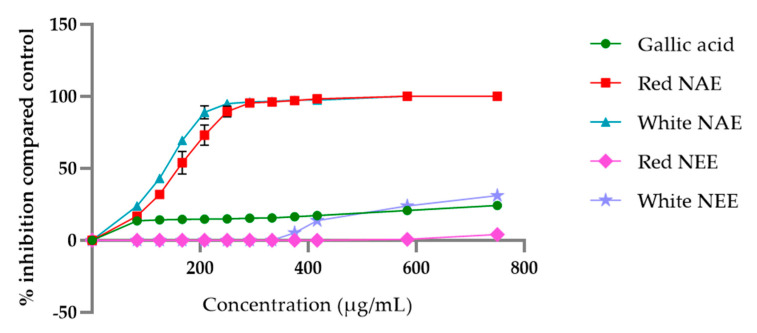
The percent of inhibition of H_2_O_2_ of both aqueous and ethanolic extracts of red *N. nucifera* petals and white *N. nucifera* petals. (NAE: *N. nucifera* petals aqueous extraction, NEE: *N. nucifera* petals ethanolic extraction). Data are mean values ± standard deviation (error bars).

**Figure 4 plants-10-01375-f004:**
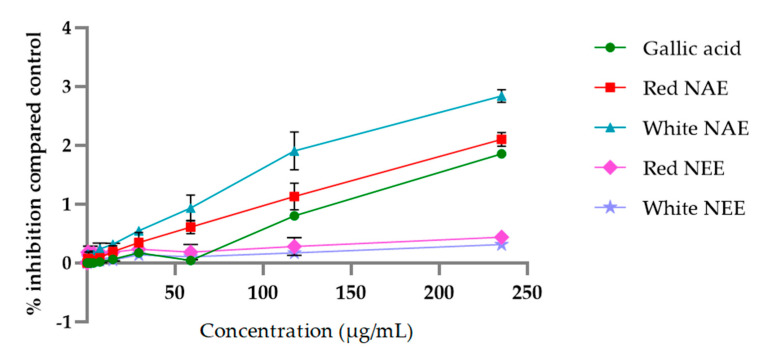
The absorbance of reducing Fe (III) to Fe (II) of both aqueous and ethanolic extracts of red *N. nucifera* petals and white *N. nucifera* petals. (NAE: *N. nucifera* petals aqueous extraction, NEE: *N*. *nucifera* petals ethanolic extraction). Data are mean values ± standard deviation (error bars).

**Figure 5 plants-10-01375-f005:**
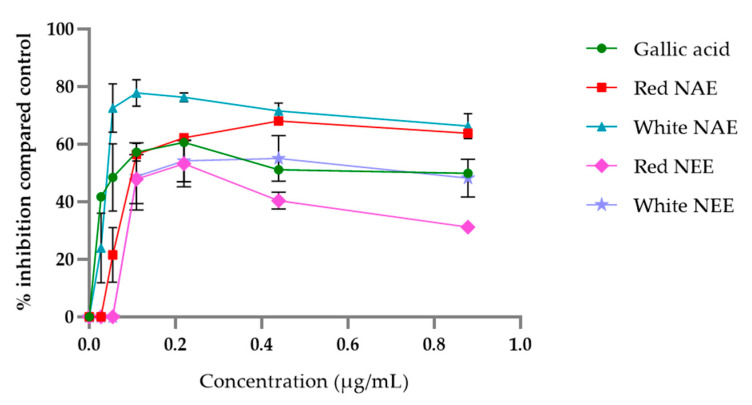
The percent of inhibition of LPO of both aqueous and ethanolic extracts of red *N. nucifera* petals and white *N. nucifera* petals. (NAE: *N. nucifera* petals aqueous extraction, NEE: *N. nucifera* petals ethanolic extraction). Data are mean values ± standard deviation (error bars).

**Figure 6 plants-10-01375-f006:**
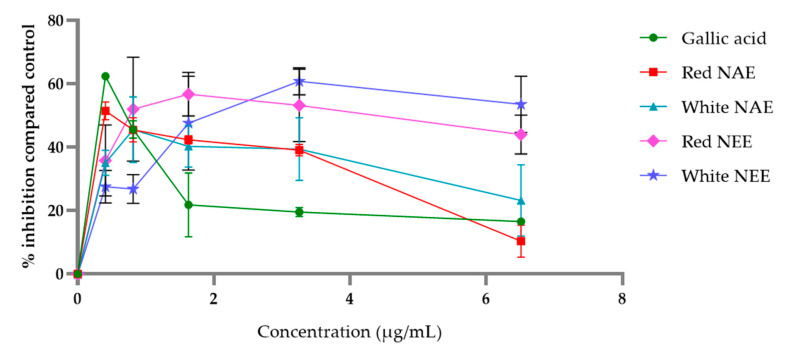
The percent of inhibition of AOPP of both aqueous and ethanolic extracts of red *N. nucifera* petals and white *N. nucifera* petals. (NAE: *N. nucifera* petals aqueous extraction, NEE: *N. nucifera* petals ethanolic extraction). Data are mean values ± standard deviation (error bars).

**Figure 7 plants-10-01375-f007:**
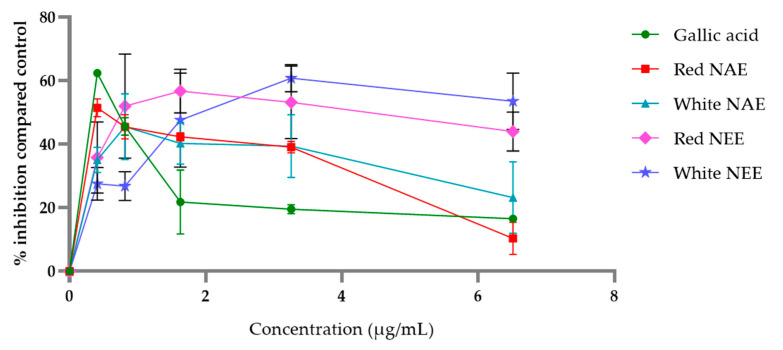
The percent of inhibition of AGEs of both aqueous and ethanolic extracts of red *N. nucifera* petals and white *N. nucifera* petals. (NAE: *N. nucifera* petals aqueous extraction, NEE: *N. nucifera* petals ethanolic extraction). Data are mean values ± standard deviation (error bars).

**Figure 8 plants-10-01375-f008:**
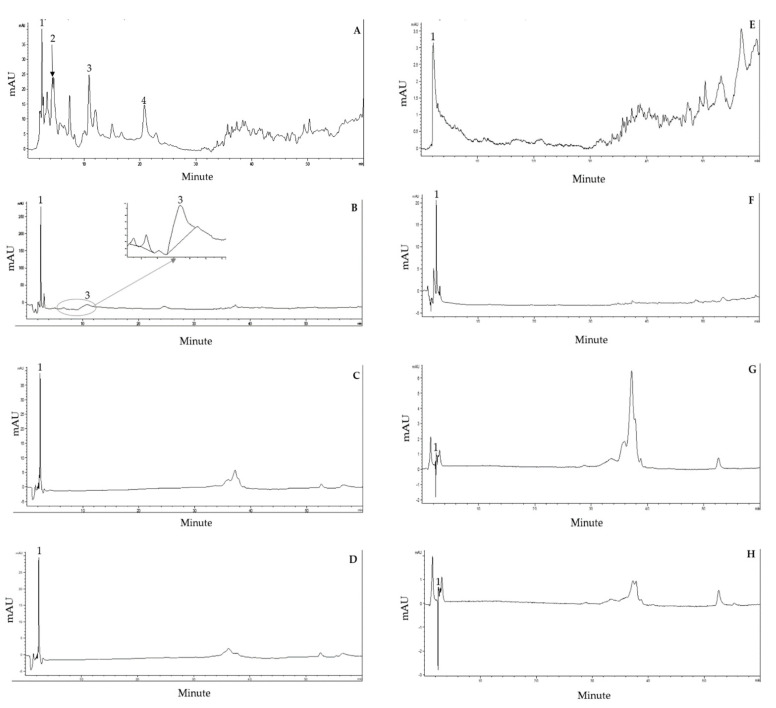
HPLC chromatograms of phenolic acids showed in the red NAE (**A**), the white NAE (**B**), the red NEE (**C**), and the white NEE (**D**) from Column, Purospher^®^ Star PR-18; mobile phase, 0.1% formic acid in water and CAN; flow rate, 0.8 mL/min; detection wavelength, 250 nm. Flavonoids presented in the red NAE (**E**), the white NAE (**F**), the red NEE (**G**) and the white NEE (**H**); detection wavelength, 330 nm. Peak identification: peak 1, quercetin; peak 2, gallic acid; peak 3, catechin; peak 4, *p*-hydroxybenzoic acid.

**Figure 9 plants-10-01375-f009:**
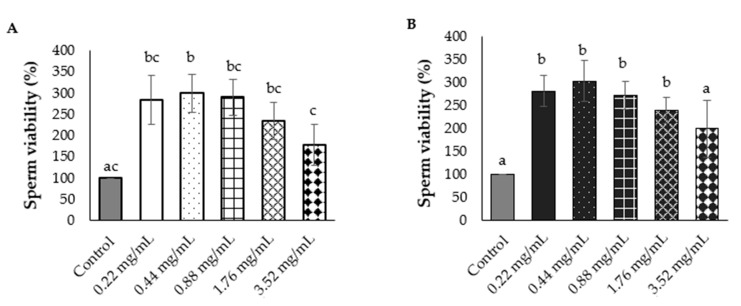
The percent of sperm viability of the aqueous extract of red *N. nucifera* petals (**A**) and white *N. nucifera* petals (**B**) compared with the control group. ^a,b,c^ The variables with the different letters indicate significant differences between groups at *p* < 0.05 (One-way ANOVA followed by Tukey’s test). The variables with the same letter are not statistically significant. Data are mean values ± standard deviation (error bars).

**Figure 10 plants-10-01375-f010:**
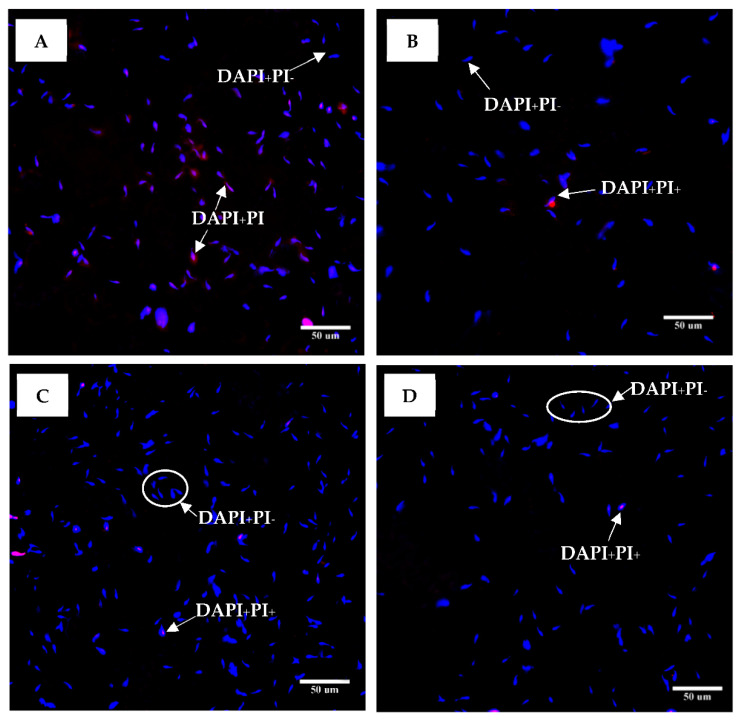
Sperm viability classification in the control group (**A**,**B**), the red NAE 0.44 mg/mL of (**C**) and the white NAE 0.44 mg/mL (**D**) stained with DAPI (4′,6-Diamidino-2-Phenylindole, Dihydrochloride) and PI (Propidium iodide). The sperm was stained blue with DAPI and stained red with PI (DAPI+PI+) was considered dead, but sperm that was stained blue with DAPI and remained unstained with PI (DAPI+PI-) was viable spermatozoa. The investigated sperm was photographed at a magnification of 400× under ImageXpress Micro 4 High-Content Imaging System.

**Figure 11 plants-10-01375-f011:**
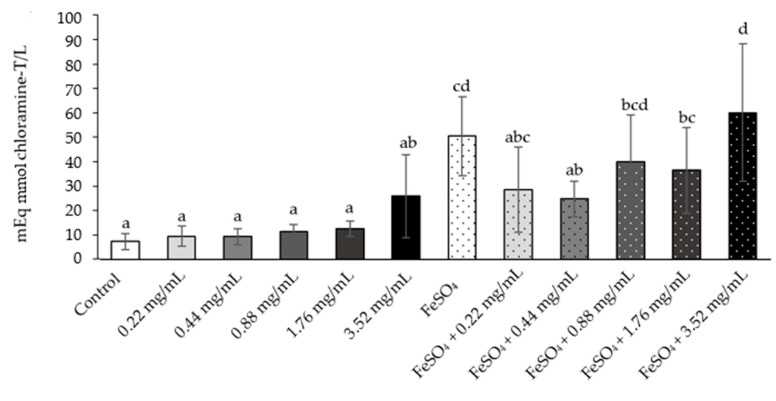
The mean value of AOPP formation in the rat sperm sample treated as follows: control, white NAE 0.22, 0.44, 0.88, 1.76 and 3.52 mg/mL, FeSO_4_ and FeSO_4_ following the white NAE 0.22, 0.44, 0.88, 1.76 and 3.52 mg/mL (NAE: *N. nucifera* petals aqueous extraction). ^a,b,c,d^ The variables with the different letters indicate significant differences between groups at *p* < 0.05. The variables with the same letter are not statistically significant. Data are mean values ± standard deviation (error bars).

**Figure 12 plants-10-01375-f012:**
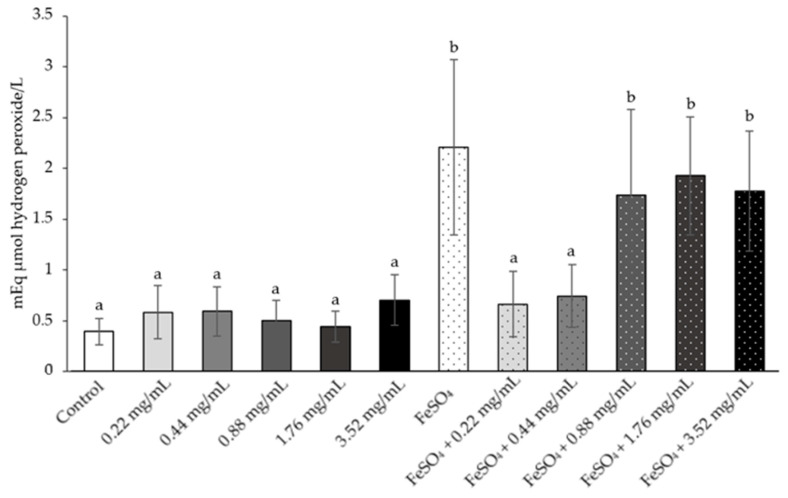
The mean value of the total oxidative status on the rat sperm sample treated as follows: control, white NAE 0.22, 0.44, 0.88, 1.76 and 3.52 mg/mL, FeSO_4_ and FeSO_4_ following the white NAE 0.22, 0.44, 0.88, 1.76 and 3.52 mg/mL (NAE: *N. nucifera* petals aqueous extraction). ^a,b^ The variables with the different letters indicate significant differences between groups at *p* < 0.05. The variables with the same letter are not statistically significant. Data are mean values ± standard deviation (error bars).

**Figure 13 plants-10-01375-f013:**
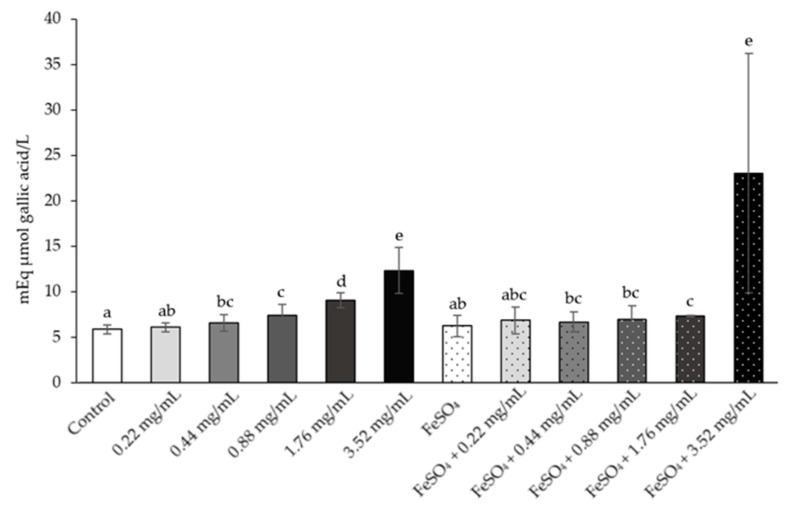
The mean value of the total antioxidant status on the rat sperm sample treated as follows: control, white NAE 0.22, 0.44, 0.88, 1.76 and 3.52 mg/mL, FeSO_4_ and FeSO_4_ following the white NAE 0.22, 0.44, 0.88, 1.76 and 3.52 mg/mL (NAE: *N. nucifera* petals aqueous extraction). ^a,b,c,d,e^ The variables with the different letters indicate significant differences between groups at *p* < 0.05. The variables with the same letters are not statistically significant. Data are mean values ± standard deviation (error bars).

**Figure 14 plants-10-01375-f014:**
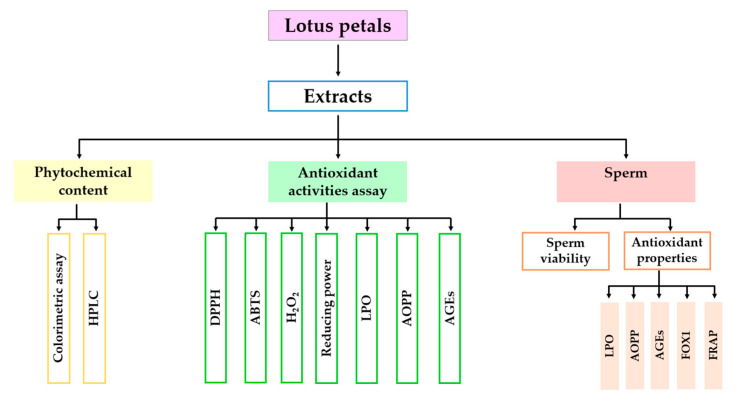
A schematic diagram showing course of research activities.

**Table 1 plants-10-01375-t001:** The total phenolic, total tannins, total flavonoid, total monomeric anthocyanins and lycopene content of red and white *N. nucifera* petal at different method of extraction.

Sample	Extraction	Total Phenolic (µg GAE/g Dried Weight)	Total Tannins (µg TAE/g Dried Weight)	Total Flavonoid (µg QE/g Dried Weight)	Total Monomeric Anthocyanins (µg Cyanidin-3-Glucoside E/g Dried Weight)
**Red**	Aqueous	20.67 ± 1.65 ^a^	17.16 ± 3.01 ^a^	10.61 ± 1.25 ^a^	49.75±11.42 ^a^
	95% ethanol	0.37 ± 0.44 ^b^	1.05 ± 0.21 ^b^	19.74 ± 0.77 ^b^	ND ^b^
**White**	Aqueous	24.15 ± 1.32 ^c^	20.52 ± 1.07 ^c^	7.83 ± 0.37 ^c^	ND ^b^
	95% ethanol	0.67 ± 0.45 ^b^	1.43 ± 0.49 ^b^	21.59 ± 1.00 ^d^	ND ^b^

^a,b,c,d^ The variables with the different letters indicate significant differences between groups in column data at *p* < 0.05 (Mann-Whitney tests were performed to analyze the differences between groups). The variables with the same letter are not statistically significant.

**Table 2 plants-10-01375-t002:** The half-maximal inhibitory concentration (IC50) values of the DPPH, ABTS, H_2_O_2_ scavenging activity and reducing power activity of gallic acid, red and white *N. nucifera* petals with different methods of extraction.

Sample	Extraction	IC50 (µg/mL)			
		DPPH	ABTS	H_2_O_2_	Reducing Power
**Red *N. nucifera***	Aqueous	14.60 ± 1.55 ^a^	13.47 ± 2.58 ^a^	160.38 ± 6.72 ^a^	90.84 ± 12.34 ^a^
	95% ethanol	634.79 ± 21.65 ^b^	173.89 ± 25.52 ^b^	1391.22 ± 62.99 ^b^	398.75 ± 43.13 ^b^
**White *N. nucifera***	Aqueous	13.31 ± 1.96 ^a^	10.95 ± 0.56 ^a^	134.02 ± 2.29 ^c^	59.30 ± 11.08 ^c^
	95% ethanol	481.41 ± 18.53 ^c^	127.45 ± 22.84 ^c^	1014.98 ± 16.94 ^d^	483.85 ± 66.38 ^b^
**Gallic acid**		14.95 ± 0.18 ^a^	0.45 ± 0.03 ^d^	1965.88 ± 4.48 ^e^	224.91 ± 3.14 ^e^

^a,b,c,d,e^ The variables with the different letters indicate significant differences between groups in column data at *p* < 0.05 (DPPH and ABTS analyzed by One-way ANOVA followed by Tukey’s test; H_2_O_2_ and reducing power were analyzed using Mann-Whitney tests to analyze the differences between groups). The variables with the same letter are not statistically significant.

**Table 3 plants-10-01375-t003:** The half-maximal inhibitory concentration (IC50) values of lipid peroxidation, AOPP and AGEs of gallic acid, red and white *N. nucifera* petal with different methods of extraction.

Sample	Extraction	IC50 (µg/mL)		
		LPO	AOPP	AGEs
**Red**	Aqueous	0.10 ± 0.00 ^a^	2.66 ± 0.49 ^a^	0.55 ± 0.24
	95% ethanol	0.17 ± 0.10 ^a^	11.88 ± 0.80 ^b^	1.00 ± 0.70
**White**	Aqueous	0.05 ± 0.01 ^b^	0.35 ± 0.07 ^c^	1.07 ± 0.93
	95% ethanol	1.35 ± 0.60 ^c^	27.19 ± 6.00 ^d^	1.23 ± 1.35
**Gallic acid**		0.11 ± 0.01 ^a^	4.21 ± 0.07 ^e^	1.05 ± 0.12

^a,b,c,d,e^ The variables with the different letters indicate significant differences between groups in column data at *p* < 0.05 (LPO and AGEs analyzed by One-way ANOVA followed by Tukey’s test; AOPP were analyzed using Mann-Whitney tests to analyze the differences between groups). The variables with the same letter are not statistically significant.

## Data Availability

The authors declare that the data supporting the findings of this study are available with in the article.

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
