# Peer review of "Phytochemical Screening, Antioxidant and Sperm Viability of Nelumbo nucifera Petal Extracts"

_plants, 2021, doi:10.3390/plants10071375_

Round 1

Reviewer 1 Report

The authors of the manuscript presented for review should address the following issues, before the work can be considered for publication:

  1. please use the full binomial name of the plant material on first apparition in text.
  2. the introduction should be enhanced with supplementary literature data regarding the potential application of N. nucifera, in connection with the aspects studied in the present work.
  3. please explain the extract obtaining methodology. I cannot understand the phrase "The extract was filtered and diluted with extract solution before experimentation."  
  4. in the Materials and methods chapter introduce the analytical equipment used in the study
  5. Conclusions chapter should be corrected. What does the authors want to present by "The specific mechanisms of bioactive compounds,the used of white NAE in animals and human as a nutrition supplement and male reproduction should be further studies." 
  6. overall, the manuscript needs an extensive language and spelling check
  7. the discussion chapter should compare the results obtained in the study with relevant literature data.

Reviewer 2 Report

In the present study the authors analyze the phytochemical components (tannins, phenols, anthocyanins, and lycopene) of two extracts (aqueous and ethanolic) of both red and white N. nucifera petals. The authors also investigated the antioxidant properties of the extracts and potential effects on sperm viability. They conclude that the white aqueous extract had the highest total phenolics content, total tannins content and maximal antioxidant activity; these characteristic increased the sperm viability in in vitro model.

The use of natural compounds to prevent or treat infertility is currently eliciting considerable interest. The paper is well written and well-structured; the presentation is clear and the language is fluent and precise. This reviewer has some comments:

  1. It has been reported that some polyphenols such as quercetin will have a beneficial or detrimental effect on cell survival depending on the cellular redox state, the flavonoid intracellular concentration and its subcellular localization. Moreover, despite quercetin has an antioxidant effect, it has been shown that quercetin produces superoxide anion by auto-oxidation which leads to the formation of hydrogen peroxide. Did the authors evaluate if their extracts have this pro-oxidant behavior?
  2. The letters to indicate statistical significance in the figures are a little confuse. I suggest that the authors should explain if the variables with the same letter are not statistical significant and for the variables with the different letter the difference is statistically significant.
  3. There are some typing errors that should be corrected.

Reviewer 3 Report

The paper describes an example of innovative use of agriculture waste, where lotus (Nelumbo nucifera) petal is presented as a source of valuable phenolic compounds with antioxidant properties. Lotus petal was extracted with water and ethanol. The aqueous (nae) and ethanolic (nee) lotus petal extracts were semi-quantitatively analyzed for total phenols, total flavonoids, total tannins, total anthocyanins, and lycopene content. NAEs were further qualitatively analyzed with HPLC. The extracts were tested for antioxidant properties with various assays and their influence on sperm viability was measured in vitro. The manuscript could be better organized. The methodological approach is comprehensive, a lot of experimental work was done. However, I suggest to add a simple schematic diagram showing course of research activities that were done. This would help a reader to follow “the story”. The research is placed in the context in a clear and exact way. The title of the paper in my opinion could be clearer and more informative. Abstract and introduction are correctly written. The results are presented in a clear way, however certain graphs/chromatograms are hard to read and therefore need to be improved. The part describing results could be shorter. Some parts in the results are somehow repeating and therefore this part is harder to read.  Generally, the style of writing is clear and adequate to me, however some parts are unclear. The manuscript needs to be reviewed by a native English speaker. Terminology is correct. The methods used are adequate. Novelty is suitable, the information is new. The article is appropriate for publication after considering all the comments and suggestions. Other comments are listed below.

P1, L14: I suggest as first to use a common name of the plant and then binomial nomenclature (in brackets). Please revise.

P1, L15-16. Unclear sentence. Revise it.

P1, L24-25. Again, the sentence does not sound clear to me. Please revise.

P1, L26. Totally agree with the message. Such like valorization would increase the value of agricultural waste significantly.

P1, L30. Same as above. The binomial nomenclature is duplicated here. I suggest the binomial name is equipped by trivial/common name of the plant.

P1, L34-36. Please explain the sentence. Make it clear also for a reader not familiar with the research field.

P2, L47. Not clear.

p2, L51. Typo.

P2, L58. And these significant side effects are ...? Please complement.

P2, L58-60. Natural antioxidants were extracted, purified and isolated form plants. The antioxidants mentioned are natural products of plants. Please revise this part.

P2, L68-69. You have already mentioned this in the material and methods. Delete this sentence.

P2, 74-77. The sentence is confused and unclear. Please correct it.

P2, L85-86. Same as above, you have already mentioned this. Delete the sentence. Here in the results, authors should avoid such writing. The results should be presented to a reader in a clear and direct way.

P3, L114-115. The graphical presentation is not clear. Graphs are difficult to read. I suggest to clearly separate between the red and white NE by choosing different colors and markers per each group of the extracts. Please correct.

P6, L149-151. It is obvious that a sugar unit of anthocyanin increase the solubility of these molecules in water.

P8, L183-185. And? Where do you have the proof that the peaks were adequately separated? Please describe the hplc results more precisely, provide the retention times, show the comparison of uv spectra between separated compound and a reference. Such like writing leaves a reader "cold".

P9, L195. HPLC trace of red NAE (A). Which peak is referred as to gallic acid? Actually number 2 is placed above two peaks that are not sufficiently separated among each other. Use an arrow to clearly show on a peak which was identified as gallic acid.

P9, L195. HPLC of a white NAE (B). A peak for catechin cannot be seen on this chromatogram. You can only see shoulder of the base line. Please correct.

P9, L197-200. Since ethanol gave a large amount of total flavonoids, it would be great to see also hplc of red and white NEE. If possible, please add in the paper two chromatograms for ethanol NE.

P13, L268-269. I'm not sure if that is correct statement. Sugars are also extracted with water from plant material. Reducing sugars are generally known to react with fc reagent, and can skew the results of TP. Ethanol has been reported as suitable solvent for extraction of phenolic compound. Anyhow, please revise this part.

P13, L269-272. As mentioned above. Water can extract also other compounds that can react with fc reagent. Generally, ethanol, methanol or acetone are known to be strong solvents for extraction of phenolic compounds. Results of HPLC of ethanol NE would be interesting thing to see, these would be useful here.

P13, L266-298. In my opinion, the discussion on phenolic compounds, their extraction with less- or more-polar solvents is not comprehensive developed. The results should be discussed in more exact way. There are many researches reports available dealing with this topic. Please revise this part.

P14, L352. Please add coordinates of the location of sampling.

P14. L356-359. Provide more info on extraction. Which extraction technique was used, time of extraction, how many times were extracts diluted, the concentration of final solutions? Please revise.

P14, L363. Maybe I have somehow missed this, but should the supplier of the reagents be mentioned here? Anyhow, please follow the journal guidelines.

P15, L374-379. You have used the same reagents as for the total phenols. Could you please add a sentence or two explaining the main/key difference of measuring total tannins and total phenols, respectively, using these reagents (fc and na2co3)?

P17, L468. Please explain, why only the NAE were analyzed with hplc. Why did you choose not to analyze NEE? Please add a sentence or two explaining this.

P17, L474-475. How were the hplc peaks identified? This is not clearly described. Did you used external standards? Comparison of retention times and spectra of separated compounds to those of chromatographic standards? please revise.

P17, L500. Briefly describe how the microscopy was done and at what conditions the micrographs were taken. It is also not clear, how the results are actually presented here.

P18, L21. Provide the data that confirms a normal distribution of measurements.

P19, L570. In-text citations, the reference list, and graphical material and tables have to be designed and organized according to the journal guidelines.

Round 2

Reviewer 1 Report

The authors addressed in satisfactory manner  the points raised by the reviewer. In my opinion, the manuscript can be accepted in the present form. 

Author Response

Dear Reviewer

Thank you very much for your “accepting our manuscript in the present form for publication in Special Issue “Antioxidant Activity of Medical Plants” in the article Manuscript ID “Plants-1266863”.

Kind regards